# Generalization in Generative Adversarial Networks: A Novel Perspective from Privacy Protection

**Bingzhe Wu[1], Shiwan Zhao[2], ChaoChao Chen[3], Haoyang Xu[1]**
**Li Wang[3], Xiaolu Zhang[3], Guangyu Sun[1,4]\*, Jun Zhou[3]**
[1]Peking University, [2]IBM Research, [3]Ant Financial,
[4] Advanced Institute of Information Technology, Peking University
{wubingzhe, xuhaoyang, gsun}@pku.edu.cn
{zhaosw}@cn.ibm.com
{chaochao.ccc, aymond.wangl, yueyin.zxl, jun.zhoujun}@antfin.com

## Abstract

In this paper, we aim to understand the generalization properties of generative adversarial networks (GANs) from a new perspective of privacy protection. Theoretically, we prove that a differentially private learning algorithm used for training the GAN does not overfit to a certain degree, i.e., the generalization gap can be bounded. Moreover, some recent works, such as the Bayesian GAN, can be reinterpreted based on our theoretical insight from privacy protection. Quantitatively, to evaluate the information leakage of well-trained GAN models, we perform various membership attacks on these models. The results show that previous Lipschitz regularization techniques are effective in not only reducing the generalization gap but also alleviating the information leakage of the training dataset.

## 1 Introduction

In the past years, generative adversarial networks (GANs) [12] have achieved remarkable progress in a wide range of applications including image translation [41, 17], image manipulation [40, 6], and image super-resolution [37, 21], etc. More recently, numerous advanced techniques [1, 22, 23, 39] are proposed for improving and stabilizing the training of GANs, leading to more realistic generated images.

Despite the tremendous success, there are still numerous open problems to be fully solved, ranging from the theoretical analysis of different regularization techniques [23, 29] to the visualization of different objective functions of GANs [3]. Among these problems, a critical one is how to formally characterize the generalization ability of GANs. Some recent studies attempted to explore this problem in different contexts. For instance, a seminal work in this direction [2] proposed the neural net distance, and the authors further showed the generalization properties of this distance. Qi et al. [26] were motivated by the progress in the Lipschitz regularization and proposed a loss-sensitive GAN. They then developed the Lipschitz regularization theory to analyze the generalization ability of the loss-sensitive GAN.

Different from the prior works [26, 2], in this paper, we aim to study the generalization ability of GANs in a relatively general setting from a novel perspective of privacy protection. Our study is motivated by a well-known intuition [38], *"reducing the generalization gap"* and *"protecting the individual's privacy"* share the same goal of encouraging a neural network to learn the population's features instead of memorizing the features of each individual, i.e., the smaller the generalization gap

is, the less information of the training dataset will be revealed. The goal of this paper is to validate this natural intuition theoretically and quantitatively.

In the theoretical side, we leverage the stability-based theory [31] to bridge the gap between differential privacy [11] and the generalization, i.e., a differentially private learning algorithm does not overfit to a certain degree. Based on the theoretical analysis, we also provide a new perspective from privacy protection to understand a number of recent techniques for improving the performance of GANs, e.g., various Lipschitz regularization terms [13, 23] and training GANs using Bayesian learning [28, 15].

In the experimental side, we quantitatively validate the relationship between the generalization gap and the information leakage of the training dataset. To this end, we introduce the *membership attack* [32] to evaluate the information leakage of a trained GAN model. In the context of machine learning, the membership attack refers to inferring whether a specific item is from the training dataset while given the trained model (discriminator and generator in our case). Specifically, we design different attack models and perform membership attacks on GANs trained with various objective functions and regularization terms. The results show that previous Lipschitz regularization techniques are effective in not only reducing the generalization gap but also alleviating the information leakage of the training dataset, which implicitly validates the aforementioned intuition. The results also suggest that it is possible to design new variants of GAN from the perspective of building privacy-preserving learning algorithms, which can bring significant regularization effects while provide appealing property of protecting the sensitive information contained in the training dataset.

The rest of this paper is organized as follows. We first briefly review related works in Section 2. Then we demonstrate our theoretical analysis in Section 3. Subsequently, we present the quantitative analysis in Section 4. At last, we conclude this work in Section 5.

## 2 Related Work

**Generative adversarial networks.** In the past years, starting from the illuminative work of GANs [12], many efforts have been devoted to this research area. Numerous researchers have tried to improve the performance of GANs from different perspectives [1, 26, 25, 23]. One direction is to improve the original objective function [26, 1, 25]. For example, to solve the gradient vanishing problem, the Least-square GAN [22] proposed using the least square loss function in the training of GANs. Wasserstein GAN (WGAN) [1] replaced the original Jensen–Shannon (JS) divergence with the Wasserstein distance and proposed an effective approximation method to compute the distance. Besides the improvements on the objective function, lots of algorithmic tricks have been proposed in empirical studies of training GANs. A typical direction is to add Lipschitz constraints on the discriminator, which enables the discriminator to be Lipschitz continuous with respect to the input and the weight. For instance, WGAN proposed using weight clipping to constrain the Lipschitz constant [1]. Gulrajani et al. [13] further suggested to use the gradient penalty to obtain better performance. Miyato et al. [23] took a different angle to regularize the spectra of the weight matrix which can implicitly constrain the Lipschitz constant.

Among these empirical techniques in training GANs, some researchers focus on building the theoretical framework to analyze the properties of GANs under different assumptions. Here we focus on the works of studying the generalization properties of GANs. Specifically, Arora et al. [2] argued that the objective function of the original GAN does not provide a theoretical generalization guarantee. As a result, the authors turned into analyzing the generalization gap of their proposed neural network distance. Qi et al. [26] developed a set of theoretical tools to analyze the generalization of GANs under the assumption of Lipschitz continuous of the discriminator. In practical, to meet the assumption, they designed a novel objective function to directly minimize the Lipschitz constant.

**Membership attacks towards deep learning algorithms.** Recently, membership attacks have arisen as a common threat model against machine learning algorithms and attained increasing attraction from the research community [33, 32, 5, 14]. A pioneer work [32] investigated the risk of membership attacks on different machine learning models. Specifically, they developed a shadow training technique to obtain an attack model in the black-box setting (i.e., without knowing the machine learning model structure and parameters). Carlini et al. [5] proposed a metric to measure the vulnerability of deep learning models. In the context of GANs, Hayes et al. [14] studied membership attacks against GANs in both black-box and white-box settings. In this paper, we use the membership attack to assess the information leakage of the dataset used for training GAN models.

# 3 Theoretical Analysis

In this part, we aim to bridge the gap between the privacy protection and the generalization ability of GANs. At a high level, we prove that a differentially private learning mechanism (i.e. training algorithm) does not overfit to a certain extent. The core idea is based on the stability-based theory [31]. To this end, we first introduce some basic notations in GANs. Then we present the theoretical analysis to characterize the generalization ability of GANs from the perspective of privacy protection. At last, based on our theoretical insight from privacy protection, we show a new interpretation of the previous theoretical results of uniform convergence [2], as well as the recent efforts on the Bayesian GAN.

**Preliminaries.** The GAN framework consists of a generator and a discriminator. We denote $\mathcal{H}_G$ and $\mathcal{H}_D$ as the hypothesis spaces of the generator and discriminator, respectively. In practice, we make use of neural networks to build $\mathcal{H}_G$ and $\mathcal{H}_D$. Formally, we have $\mathcal{H}_G = \{\mathbf{g}(z; \theta_g), \theta_g \in \mathbb{R}^p\}$ and $\mathcal{H}_D = \{\mathbf{d}(x; \theta_d), \theta_d \in \mathbb{R}^q\}$, where $\mathbf{g}$ and $\mathbf{d}$ are multi-layer convolutional neural networks. $\theta_g$ and $\theta_d$ are the corresponding weight parameters. The training of GANs can be seen as playing a min-max game to solve the following optimization problem:

$$\min_{\theta_g} \max_{\theta_d} \mathbb{E}_{x \sim p_{data}}[\phi(\mathbf{d}(x; \theta_d))] + \mathbb{E}_{z \sim p_z}[\phi(1 - \mathbf{d}(\mathbf{g}(z; \theta_g); \theta_d))] \tag{1}$$

The above formulation can be seen as an extended version of the objective function used in the vanilla GAN [12]. According to previous literature [2], we call function $\phi$ the *measuring function*. Note that setting $\phi(t) = \log(t)$ leads to the objective function used in the work of the original GAN [12], while the recent WGAN [1] proposed using $\phi(t) = t$.

To optimize Equation 1, we need to build two learning mechanisms $\mathcal{A}_d$ and $\mathcal{A}_g$. During the training, we alternately run $\mathcal{A}_d$ and $\mathcal{A}_g$ to seek an equilibrium $(\theta_d^*, \theta_g^*)$. Specifically, $\mathcal{A}_d$ tries to find $\theta_d^*$ such that it maximizes the expected loss of the discriminator:

$$U(\theta_d, \theta_g^*) = \mathbb{E}_{x \sim p_{data}}[\phi(\mathbf{d}(x; \theta_d))] + \mathbb{E}_{z \sim p_z}[\phi(1 - \mathbf{d}(\mathbf{g}(z; \theta_g^*); \theta_d))] \tag{2}$$

and $\mathcal{A}_g$ tries to find $\theta_g^*$ to minimize the expected loss of the generator:

$$V(\theta_d^*, \theta_g) = \mathbb{E}_{z \sim p_z}[\phi(1 - \mathbf{d}(\mathbf{g}(z; \theta_g); \theta_d^*))] \tag{3}$$

**Empirical loss and generalization.** The optimization of Equation 2 and 3 can not be directly solved since the expectation over the distribution of the true data $p_{data}$ is intractable. Instead, we approximate them with empirical loss on a set of i.i.d. real data samples $S = \{x_1, x_2, \cdots, x_m\}$ and noise vectors $Z = \{z_1, z_2, \cdots, z_m\}$ drawn from $p_{data}$ and $p_z$, respectively[2]. We denote the resulted empirical versions of Equation 2 and 3 as $\hat{U}$ and $\hat{V}$. In this empirical setting, the learning mechanisms $\mathcal{A}_d$ and $\mathcal{A}_g$ turn into the role as the empirical loss optimizers, which are to optimize the empirical loss $\hat{U}$ and $\hat{V}$, i.e., finding $(\hat{\theta}_d^*, \hat{\theta}_g^*)$. To study the generalization property of the learning mechanisms, we need to evaluate the generalization gap between the empirical and expected objective losses. In this paper, we mainly focus on the analysis of Equation 2 since our viewpoint is from the privacy protection and Equation 3 does not explicitly touch the original training data. As a common practice shown in the prior work [12], we analyze Equation 2 when $\theta_g^*$ is given. Formally, we define the generalization gap as follows (we take the discriminator loss $U$ as an example):

$$F_U(\mathcal{A}_d) = \mathbb{E}_{\theta_d \sim \mathcal{A}_d(S)} \mathbb{E}_{S \sim p_{data}^m}[\hat{U}(\theta_d, \theta_g^*) - U(\theta_d, \theta_g^*)] \tag{4}$$

where $S \sim p_{data}^m$ denotes sampling $m$ training samples from the oracle distribution $p_{data}$. In the above equation, we take the expectation with respect to the randomness in the learning mechanism and also in the sampling process similar to the literature [24, 36]. Note that we can infinitely sample from $p_z$ (e.g. a uniform distribution) and $p_z$ is irrelevant to the original training data (i.e. sampling from $p_z$ does not induce the leakage of the training dataset). As a result, we omit the second term in the RHS of Equation 2 and focus on studying the first term.

**Privacy protection and generalization bound.** To bridge the gap between privacy protection and the generalization bound, we need to characterize how an algorithm can protect privacy, i.e., the amount of information leakage of the training dataset. Differential privacy [10] is seen as a gold standard for privacy protection in the security community. It provides a rigorous bound on privacy

cost of the algorithm, even in the worst case. The definition of differential privacy is based on the adjacent datasets. Two datasets are adjacent when they differ in a single element. Then we can introduce differential privacy as follows:

**Define 1** (*Differential privacy*) *A randomized algorithm* $\mathcal{A} : D \rightarrow R$ *satisfies* $\epsilon$-*differential privacy if for any two adjacent datasets* $\mathcal{S}, \mathcal{S}^{'} \subseteq D$ *and for any subset of outputs* $O \subseteq R$ *it holds:*

$$\mathbf{Pr}[\mathcal{A}(\mathcal{S}) \in O] \leq e^{\epsilon}\mathbf{Pr}[\mathcal{A}(\mathcal{S}^{'}) \in O] \tag{5}$$

In our setting, $\mathcal{A}$ can be the training algorithm (i.e. $\mathcal{A}_d$). Intuitively, Equation 5 indicates that participation of one individual sample in the training phase has a negligible effect on the final weight parameters. A relevant concept is *uniform RO-stable* of an algorithm. An algorithm is stable if a small change to the input causes a limited change in the output. Here, RO denotes *"replace one element in the input"*. The above description is made formally as:

**Define 2** (*Uniform RO-stability*) *The randomized algorithm* $\mathcal{A}$ *is uniform RO-stable with respect to the discriminator loss function (Equation 2) in our case, if for all adjacent datasets* $S, S'$, *it holds that:*

$$\sup_{x \in S} |\mathbb{E}_{\theta_d \sim \mathcal{A}(S)}[\phi(\mathbf{d}(x; \theta_d))] - \mathbb{E}_{\theta_d \sim \mathcal{A}(S')}[\phi(\mathbf{d}(x; \theta_d))]| \leq \epsilon_{stable}(m) \tag{6}$$

A well-known heuristic observation is that differential privacy implies uniform stability. The prior work [36] has formlized this observation into the following lemma:

**Lemma 1** (*Differential privacy* $\Rightarrow$ *uniform RO-stability*) *If a randomized algorithm* $\mathcal{A}$ *is* $\epsilon$-*differentially private, then the algorithm* $\mathcal{A}$ *satisfies* $(e^{\epsilon} - 1)$-*RO-stability.*

The stability of the algorithm is also related to the generalization gap. Numerous studies [31, 24] focus on exploring the relationship in various settings. Formally, we have the following lemma:

**Lemma 2** *If an algorithm* $\mathcal{A}$ *is uniform RO-stable with rate* $\epsilon_{stable}(m)$, *then* $|F_U(\mathcal{A})|$ *(Equation 4) can be bounded:* $|F_U(\mathcal{A})| \leq \epsilon_{stable}(m)$.

Intuitively, the more stable the algorithm is, the better its generalization ability will be. We take a further step to build the connection between differential privacy and the generalization gap. This can be done via combining the above two lemmas. Formally, we introduce Theorem 1 as follows:

**Theorem 1** (*Generalization gap*) *If an algorithm* $\mathcal{A}$ *satisfies* $\epsilon$-*differential privacy, then the generalization gap can be bounded by a data-independent constant.*

The proof can be accomplished by following the roadmap: $Differential\ privacy \Rightarrow Stability \Rightarrow Generalization$. The proof details can be found in Appendix. By applying Theorem 1 to $\mathcal{A}_d$, we can show that the generalizability of the discriminator is ensured when the training algorithm satisfies $\epsilon$-differential privacy. Note that we focus on the generalization of the discriminator loss, since the optimization of Equation 3 does not touch the original data. We can derive the similar generalization bound of the generator by leveraging the post-processing property of the differential privacy protocol [11], with the help of the robust generalization notations in the adaptive learning paradigm [7].

Theorem 1 not only enables characterizing GAN's generalizability from the viewpoint of privacy protection, but also helps to understand previous techniques for improving the generalization of GANs. A typical example is the Lipschitz regularization technique. Previous studies propose implementing the regularization from various angles [1, 26, 23]. For instance, the loss-sensitive GAN [26] designed a novel objective loss to restrict the discriminator to satisfy the Lipschitz condition. Spectral normalization [23] explored this direction by adding regularization on the weight parameters. And WGAN [1] proposed using gradient penalty to constrain the magnitude of the gradient, which further implicitly led to the Lipschitz condition. From the perspective of differential privacy, the Lipschitz condition for the outputs is also a crucial ingredient for building a differentially private algorithm. We also infer that adding Lipschitz constraints implicitly leads to stability of the algorithm, which can further be used for reducing the generalization gap (see more details in the evaluation section).

Above analysis focuses on the discriminator loss $U$. As mentioned above, it is natural to extend the analysis to the generator loss since the optimization of the generator loss does not touch the original training dataset and we can leverage the post-processing property of the differential privacy. We can further study the whole optimization procedure of the GAN, i.e., the alternative running of $\mathcal{A}_d$ and $\mathcal{A}_g$. These can be accomplished by the composition theory in adaptive learning theory [8], we mark this as the future work.

**Revisiting previous results on uniform convergence.** Previous works attempted to explore the uniform convergence with respect to different objective functions. For example, Qi et al. [26] proposed the loss-sensitive GAN and proved the uniform convergence with respect to the discriminator loss. Arora et al. proposed the neural distance to analyze the generalization property (uniform convergence) of GANs. Note that both of them focus on the special form of the GAN or the objective function. In this paper, based on the aforementioned theoretical results, we can prove the uniform convergence of the discriminator loss when the training algorithm satisfies the differential privacy protocol. Formally, we have the following theorem:

**Theorem 2** (***Uuniversal Bound*** ): *Suppose $\mathcal{A}_d$ satisfies $\epsilon$-differential privacy and $\mathbf{d}^{(k)}(x; \theta_d^{(k)})$ be the output of $\mathcal{A}_d$ at the $k$-th iteration. Then, $\forall k$, the generalization gap with respect to $\mathbf{d}^k$ can be bounded by a universal constant which is related to $\epsilon$.*

The proof of the above theorem can be done via combing the post-processing property [11] of differential privacy and McDiarmid's inequality [34]. The details can be found in Appendix.

**Connection to Bayesian GAN.** Recently, training GANs using Bayesian learning has emerged as a new way to avoid mode collapse [28]. A well-known interpretation of mode collapse is that the generator/discriminator has memorized some examples from the training dataset. Hence, the memorization phenomenon can also breach the privacy of individuals in the training dataset. Thus, we infer that the effectiveness of the Bayesian GAN may come from preventing information leakage of the training dataset. In what follows, we briefly introduce how we validate this conjecture with our theoretical results. Specifically, we take a recent work [28] as an example. In the work, the authors proposed using stochastic Hamiltonian Monte Carlo (HMC) to marginalize the posterior distribution of the weight parameters of both generator and discriminator. We have noted that Wang et al. [35] pointed out that sampling one element from a posterior distribution can implicitly satisfy differential privacy. Based on their results, we can prove that the HMC sampling also preserves differential privacy with minor modification (refers to Section 4 in [35]), and then the Bayesian GAN can implicitly preserve the differential privacy. Thus we are not surprised that the Bayesian GAN can alleviate mode collapse in GANs since its connection to differential privacy mentioned above.

## 4   Quantitative Analysis

In this section, we quantitatively validate the connection between the generalizability and the privacy protection/information leakage of GANs by investigating some of the most popular GAN models. In the theoretical analysis, we focus on the differentially private learning algorithms. While in practice, differential privacy is a strict requirement for most of the existing GANs so that we focus on studying information leakage of GANs instead. In particular, we note that adding Lipschitz constraints on the discriminator has recently emerged as an effective solution for improving the generalization ability of such GANs, thus we aim to study the effects of various regularization techniques for adding the Lipschitz constraints. In a nutshell, our results show that the Lipschitz constraints not only reduce the generalization gap but also make the trained model resistant to the membership attacks which are used for detecting the sensitive information contained in the training dataset.

This section is structured as follows. First, we will introduce the experimental settings including the datasets and the choice of different hyper-parameters. Then, we demonstrate our main results on different datasets. At last, we provide some discussions on the attack methods and the regularization techniques.

### 4.1   Experimental Setup

**Datasets.** We conduct experiments on a face image dataset and a real clinical dataset, namely, Labeled Faces in the Wild (LFW) [20] which consists of 13,233 face images, and the IDC dataset

which is publicly available for invasive ductal carcinoma (IDC) classification[3] and contains 277,524 patches of $50 \times 50$ pixels (198,738 IDC-negative and 78,786 IDC-positive).

**Model setup.** Note that we focus on studying the effects of different regularization techniques instead of the architecture design of the GAN model, thus we use the same generator architecture and the same discriminator architecture in all experiments. Specifically, we adopt DCGAN following most of previous works [27]. More details can be found in the work [27]. We set the size of the generated image to 64x64 for the LFW dataset and 32x32 for the IDC dataset, respectively. As for optimization, we use Adam [18] in all experiments, and use different hyper-parameters for different training strategies. To be specific, we make use of Adam for the GAN trained with JS divergence. The learning rates is set to $0.0004$ for the GAN trained without any regularization terms (original GAN [12]), while for other GANs (e.g. trained using Wasserstein distance), the learning rate is set to $0.0002$. More details of hyper-parameter settings (e.g. $\beta$ in Adam) can be found in Appendix. We trained all the models for 400 epochs on both datasets.

**Attack setup.** We make use of membership attacks for evaluating the information leakage of the GAN model. We build the attack model based on the output of the discriminator, which is a bounded function. We suppose $\mathbf{d}(x; \theta_d) \leq b$ for all $x$ in Equation 1. This assumption naturally holds in the context of GAN (e.g. $b = 1$ for the original GAN). This is also a common assumption in many previous works. Here, letting $b = 1$ suffices to all our cases. We then assume that the attacker $\mathcal{A}$ has access to the trained GAN model, i.e. the discriminator and generator. Note that the notation $\mathcal{A}$ is different from the previous ones that denote training algorithms. The goal of the attacker is to determine whether a record (an image in our case) in the attack testing dataset is from the original training dataset. Based on the above setting, the attack model proceeds as follows:

- Given the discriminator $\mathbf{d}(x; \theta_d)$ and an image from the attack testing dataset.
- $\mathcal{A}$ firstly sets a threshold $t \in (0, 1)$.
- $\mathcal{A}$ outputs 1 if $\mathbf{d}(x; \theta_d)/b \geq t$, otherwise, it outputs 0.

where the output of 1 indicates that the input image is from the training dataset.

To evaluate the performance of the attack model, we need to build the attack testing dataset. For the LFW dataset, we randomly choose 50% of images as the original training dataset to train the GAN model. We build the attack testing dataset by mixing the original training dataset and the remaining images. For the IDC dataset, we only use the positive part of the dataset. Since the data provider has already provided a partition of training and testing datasets [16] (22,383 images in the training dataset and 14,157 images in the testing dataset[4]), we directly use the original partition and build the attack testing dataset by mixing the training and testing datasets. We then treat the above attack as a binary classification model and evaluate its performance based on the F1 score and the AUC value. To compute the F1 score, we assume the attacker has obtained the average value of $\mathbf{d}(x; \theta_d)$ on the training dataset. Thus we can set the average value as the threshold $t$ and then compute the F1 score at this threshold.

## 4.2 Results on LFW and IDC Datasets

Here, we present the results of our evaluation on the GAN models trained with different strategies. We conduct extensive experiments of different settings. Specifically, we focus on three commonly used techniques for adding Lipschitz constraints, namely, weight clipping [1], gradient penalty [29], and spectral normalization [23]. For weight clipping, we set the clip interval as $[-0.01, 0.01]$ following the prior work [1]. We combine these regularization techniques with two types of objective functions, the traditional JS divergence (setting $\phi(t) = \log(t)$) and the Wasserstein distance (setting $\phi(t) = t$). As mentioned above, the performance of the attack model is measured by the F1 score and AUC value, and we use the gap between the testing and training losses to estimate the generalization gap. We also calculate the Inception score [29], which is a commonly used for assessing the quality of the generated images in previous works. The overall results are shown in Table 1.

**LFW.** We first conduct contrastive experiments on the GAN trained using JS divergence (setting $\phi(x) = \log(x)$ in Equation 1). As shown in Table 1, the plain model (trained without any regulariza-

Table 1: Evaluation results of DCGAN trained with different strategies. IS denotes the Inception score. N/A indicates that the strategy leads to failure/collapse of the training. The last row presents the Inception scores of the real data (training images of these two datasets).

| Strategy | LFW | | | | IDC | | | |
|---|---|---|---|---|---|---|---|---|
| | F1 | AUC | Gap | IS | F1 | AUC | Gap | IS |
| **-JS divergence-** | | | | | | | | |
| Original | 0.565 | 0.729 | 0.581 | 3.067 | 0.445 | 0.531 | 0.138 | 2.148 |
| Weight Clipping | 0.486 | 0.501 | 0.113 | 3.112 | 0.378 | 0.502 | 0.053 | 2.083 |
| Spectral Normalization | 0.482 | 0.506 | 0.106 | 3.104 | 0.416 | 0.508 | 0.124 | 2.207 |
| Gradient Penalty | N/A | | | | N/A | | | |
| **-Wasserstein-** | | | | | | | | |
| W/o clipping | N/A | | | | N/A | | | |
| Weight Clipping | 0.484 | 0.512 | 0.042 | 3.013 | 0.388 | 0.513 | 0.045 | 1.912 |
| Spectral Normalization | 0.515 | 0.505 | 0.017 | 3.156 | 0.415 | 0.507 | 0.013 | 2.196 |
| Gradient Penalty | 0.492 | 0.503 | 0.031 | 2.994 | 0.426 | 0.504 | 0.017 | 1.974 |
| IS (Real data) | 4.272 | | | | 3.061 | | | |

tion term) is more vulnerable than those trained with different regularizers. The plain model leaks some information of the training dataset, which results in the F1 score of $0.565$ and the AUC value of $0.729$ (greater than $0.5$), respectively. While various regularization techniques are used for training the GAN, the attack performance decreases drastically. For instance, while spectral normalization is used, the F1 score is dropped from $0.565$ to $0.482$, and the AUC value is dropped to $0.506$, which approximates to the random guess. Along with the decrease of the attack performance, we also observe a consistent decrease in the generalization gap (Gap in Table 1). For example, the gap is decreased from $0.581$ to $0.106$ while spectral normalization is used.

In addition to these metrics, we also calculate the Inception score of each model. As shown in Table 1, the Inception score of the LFW dataset (shown in the last row of Table 1) is much smaller than the commonly used benchmarks (e.g. Cifar10 [19] and ImageNet [9]). This is because that the Inception score is calculated by the model pre-trained by the images in ImageNet in which the image always contains general objects (such as animals) while the image in LFW always contains one face. We observe that the spectral normalization can generate images with higher visual qualities but obtain lower Inception score than the weight clipping strategy (see generated images in Appendix). Thus, these numerical results of Inception score indicate that Inception score is not suitable for some image generation tasks and we need to design a specific metric for a given task. Among these experiments, we also conduct the same attack experiments on GANs trained using Wasserstein distance and observe similar phenomena as shown in Table 1.

**IDC.** The images in the IDC dataset contain various tissues and all of these tissues have similar shapes and colors. As a result, performing the attack on the IDC dataset is more challenging than on other datasets which consist of some concrete objects (e.g. LFW and Cifar10 [19]). As we can see in Table 1, in all cases, the attack performance on IDC is lower than the performance on LFW in terms of the F1 score and AUC value. We also provide some quantitative analysis to interpret the performance drop in the following subsection. Despite the performance drop, we can still observe the effectiveness of different regularizers in reducing the generalization gap and information leakage. For example, with the use of spectral normalization, AUC drops from $0.531$ to $0.508$ and F1 score decreases from $0.445$ to $0.416$. Another notable thing is that training the GAN using Wasserstein distance without weight clipping can lead to the failure of training. This may be caused by the gradient explosion since there is no activation function to compress the output values of the last layer of the discriminator in Wasserstein GAN (in contrast, the original GAN used the sigmoid function for the compression purpose). For the results of Inception score, we can observe an obvious decrease from the LFW dataset to the IDC dataset. This may be caused by the difference of contents contained in the images (tissues in IDC images and faces in LFW images).

In summary, all the empirical evidence implies that previous regularizers for adding Lipschitz constraints can not only decrease the generalization gap but also reduce the information leakage of the training dataset in terms of the attack performance. Moreover, we observe that spectral normalization

achieves comparable visual quality of the generated images in contrast to the original GAN (see generated images in Appendix). This suggests that we can attempt to use this technique in various applications to trade off the information leakage and the visual quality of the generated images.

### 4.3 Discussions

**Attack performance on different datasets.** In the experiments, we observe that the attack performance may vary on different datasets. From Table 1, in all cases, the attack performance of IDC is always worse than the performance of LFW. For the original GAN model, the AUC of the attack model on LFW is 0.729 while the one for IDC is 0.531 which shows a 27.2% relative performance drop. We infer that the drop is caused by the higher similarity of the images in the IDC data. Quantitatively, the similarity can be measured by the standard deviations (per channel for an RGB image) of these two datasets. Specifically, the standard deviation of IDC is 0.085 (R-channel) while the value of LFW is 0.249 (R-channel). The IDC dataset has significantly lower standard deviation than the LFW dataset. This suggests that an individual example in the IDC dataset will be less likely to noticeably impact the final model which further constrains an adversary's ability. Note that similar evidence has been found in the prior work [32].

**Other weight normalization.** In the previous experiments, we have employed three regularizers for limiting the discriminator to be Lipschitz continuous. In addition to these regularizers, there are some other approaches to use weight normalization techniques to regularize the model training. The *original weight normalization* introduced in the work [30] is to normalize the l2 norm of each row vector in a weight matrix. Subsequently, Brock et al. [4] proposed *orthonormal regularization* on the weight matrix to stabilize the training of GANs. We also conduct attack experiments on the GANs trained with these two normalization methods. In practice, the orthonormal regularization achieves comparable performance in terms of the F1 score of the attack model (0.402 for the IDC dataset), while obtains comparable image quality compared with spectral normalization. In addition, the original weight normalization will lead to training failure (not convergence/mode collapse) in our cases. The failure may be caused by the conflict between the weight normalizing and the desire to use as many features as possible as discussed in the prior work [23].

**Black-box attack.** The aforementioned analysis is all based on the white-box attack, i.e., the model structure and weights of the discriminator and generator are exposed to the adversary. In this part, we present some discussions on the black-box attack. We consider black-box attacks where the adversary has limited auxiliary knowledge of the dataset fol-

Table 2: The results of black-box attack on the LFW dataset.

| Strategy | F1 | AUC | Gap |
|---|---|---|---|
| Original | 0.423 | 0.549 | 0.581 |
| Weight Clipping | 0.358 | 0.502 | 0.113 |
| Spectral Normalization | 0.347 | 0.497 | 0.106 |

low the prior work [14]. Specifically, for the LFW dataset, we assume the attacker has 30% images of both the training and testing datasets (marked as the auxiliary knowledge in the following part). Then we can use the auxiliary information to build a "fake" discriminator to imitate the behavior of the original discriminator (more details can be found in the prior work [14]). Once the "fake" discriminator is obtained, we can replace the original discriminator with the fake one to perform the aforementioned white-box attack. The results of the black-box attack are summarized in Table 2. Intuitively, the black-box is a more challenging task than the white-box attack. This can be observed by the decrease of the attack performance from the white-box to black-box (the AUC decreases from 0.729 to 0.549 for the original model). From Table 2, we can also observe the effectiveness of the spectral normalization for reducing the information leakage.

## 5 Conclusion

In this paper, we have shown a new perspective of privacy protection to understand the generalization properties of GANs. Specifically, we have validated the relationship between the generalization gap and the information leakage of the training dataset both theoretically and quantitatively. This new perspective can also help us to understand the effectiveness of previous techniques on Lipschitz regularizations and the Bayesian GAN. We hope our work can light the following researchers to leverage our new perspective to design GANs with better generalization ability while preventing the information leakage.

## Footnotes

\*Corresponding author. This work is supported by NSF China 61832020, NSF China 61572045, and Beijing Academy of Artificial Intelligence.

[2]In practice, we always sample different $Z$ while $S$ is fixed at each training round.

[3]http://www.andrewjanowczyk.com/use-case-6-invasive-ductal-carcinoma-idc-segmentation/

[4]They also provide a validation dataset and we did not use it in our experiments.

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
