[Supplementary Material · nips_appendix.pdf]

# A  Appendix

## A.1  Proof of Theorem 1

Before presenting the proof of Theorem 1, we first re-demonstrate the previous lemmas here.

**Lemma 1** *(**Differential privacy $\Rightarrow$ uniform RO-stability**) If a randomized algorithm $\mathcal{A}$ is $\epsilon$-differentially private, then the algorithm $\mathcal{A}$ satisfies $(e^{\epsilon} - 1)$-RO-stability.*

**Lemma 2** *If an algorithm $\mathcal{A}$ is uniform RO-stable with rate $\epsilon_{stable}(m)$, then $|F_U(\mathcal{A})|$ can be bounded: $|F_U(\mathcal{A})| \leq \epsilon_{stable}(m)$.*

Then, we briefly introduce the proof of Theorem 1.

**Theorem 1** *(**Generalization gap**) If an algorithm $\mathcal{A}$ satisfies $\epsilon$-differential privacy, then the generalization gap can be bounded by a data-independent constant.*

**Proof:** Given a differentially private algorithm $\mathcal{A}$ with a privacy cost $\epsilon$, then, according to Lemma 1, we can derive that $\mathcal{A}$ is uniform RO-stability. Then the generalization bound $|F_U(\mathcal{A})|$ satisfies the following inequality:

$$|F_U(\mathcal{A})| \leq \epsilon_{stable} \tag{1}$$

where $\epsilon_{stable}$ is the stability ratio (equals to $e^{\epsilon} - 1$).

## A.2  Proof of Theorem 2

The proof of Theorem 2 can be accomplished by combing the post-processing property of the differentially private algorithm and the McDiarmid's inequality. Thus, we first introduce the post-processing property and the McDiarmid's inequality as follows:

**Lemma 3** *(**Post-processing**) Let $\mathcal{A}$ be a randomized algorithm that is $\epsilon$-differently private. Let $f$ be an arbitrary randomized mapping. Then $f \circ \mathcal{A}$ is $\epsilon$-differentially private.*

**Lemma 4** *(**McDiarmid's inequality**) Consider independent random variables $X_1, X_2, \cdots, X_n \in \mathcal{X}$ and a mapping $f : \mathcal{X}^n \to \mathbb{R}$. If, for all $i \in 1, 2, \cdots, n$ and for all $x_1, x_2, \cdots, x_n, x_i' \in \mathcal{X}$, the function $\phi$ satisfies:*

$$|\phi(x_1, \cdots, x_{i-1}, x_i, x_{i+1}, \cdots, x_n) - \phi(x_1, \cdots, x_{i-1}, x_i', x_{i+1}, \cdots, x_n)| \leq c_i \tag{2}$$

*then:*

$$P(|\phi(X_1, \cdots, X_n)| - \mathbb{E}\phi| \geq t) \leq 2\exp(-\frac{-2t^2}{\sum c_i^2}) \tag{3}$$

**Theorem 2** *(**Uniform convergence**): Suppose $\mathcal{A}_d$ satisfies $\epsilon$-differential privacy and $\mathbf{d}^{(k)}(x; \theta_d^{(k)})$ be the output of $\mathcal{A}_d$ at the $k$-th iteration. Then, $\forall k$, the generalization gap with respect to $\mathbf{d}^k$ can be bounded by a universal constant which is related to $\epsilon$.*

**Proof:** For a training algorithm $\mathcal{A}$ which satisfies $\epsilon$-differentially private, then the function build by the output of $\mathcal{A}$ is $\epsilon$-differentially private in terms of Lemma 3. As a result, the computation of $U(\theta_d, \theta_g^*)$ is differentially private. Using Lemma 1, we can derive that $U(\theta_d, \theta_g^*)$ satisfies uniform RO-stability. We further let $\phi = U$, then based on the stability of $U$, we can meet the requirement of Lemma 4. Based on Lemma 4, we can obtained that:

$$P(|U(\theta_d, \theta_g^*) - \hat{U}(\theta_d, \theta_g^*)| \geq t) \leq 2\exp(\frac{-2t^2}{m\epsilon^2}) \tag{4}$$

where $\epsilon$ is the privacy cost and m denotes the number of samples used for computing the empirical loss $\hat{U}$.

## A.3   Hyper-parameter Setting

Here we list the details of the hyper-parameters of different experiments in Table 1.

Table 1: Details of hype-parameters of Adam optimizer.

| Strategy | Learning rate | $\beta_1$ | $\beta_2$ |
|---|---|---|---|
| **-JS divergence-** | | | |
| Original | 0.0004 | 0.5 | 0.999 |
| Weight clipping | 0.0004 | 0.5 | 0.999 |
| Spectral normalization | 0.0004 | 0.0 | 0.999 |
| Gradient penalty | 0.0004 | 0.0 | 0.999 |
| **-Wassertain Distance-** | | | |
| W/o clipping | 0.0002 | 0.5 | 0.999 |
| Weight clipping | 0.0002 | 0.5 | 0.999 |
| Spectral normalization | 0.0002 | 0.0 | 0.999 |
| Gradient penalty | 0.0002 | 0.0 | 0.999 |

## A.4   Visualization

Here we show some generated images produced by GANs, which are trained with different regularization techniques. As concluded in the paper, we can see that spectral normalization achieves comparable quality of generated images in contrast to the original GAN (even better on the LFW dataset).

Original          Clip          Spectral

Table 2: Visualization of generated images of GAN models trained with the JS-divergence. The first row is the results of the LFW dataset and the second row is the IDC dataset.