[Reviews · NeurIPS 2019]

Reviewer 1



Summary This paper provides a novel perspective to study the generalization of GANs. The author has theoretically and experimentally analyzed the connection between the information leakage and the generalization of GANs. Strengths 1. The insight to study the generalization of GAN from the view of privacy protection is very interesting. This work may motivate the following research to study the properties of GANs from the privacy aspect. 2. The theoretical analysis that employs differential privacy and the stability based theory is insightful 3. The experiment demonstrates that different Lipschitz regularization techniques can not only reduce information leakage but also improve the generalization ability of GANs. This conclusion is useful to guide the training of GANs in practice. Weakness 1. The attack used in this paper is relatively straightforward. The author should evaluate different attack methods and show the experimental results 2. The theoretical analysis has mentioned the connection between this work and the Bayesian GAN. It is better to demonstrate some evaluations of the information leakage of Bayesian GAN in the experimental section. Comment after rebuttal: The authors addressed the raised concerns. As pointed out by other reviewers, more advance of theoretical finding is desired.

Reviewer 2



This paper studies the generalization property of GAN models from an interesting perspective, which is built upon the intuition that reducing the generalization gap usually coincide with protecting privacy. To my knowledge, this angle of analysis is new to the body of GAN research. The empirical results/discussion on membership attack methods and various regularizations also enriches our understanding on the performance of GAN. That said, the pure theoretical advancement is good but not strong - The key results Thm1,2 are direct variations of differential privacy and may appear incremental rather than fundamental. Section 4 can be improved - it would be great if the authors can also provide guidlines of GAN design along the way of discussion.

Reviewer 3



Overall I think this paper raises an interesting perspective to understanding adversarial generative models. I think this paper has some value by raising the question and offering some interesting experimental results. The theory is quite standard, the authors first cite a relationship between differential privacy and RO stability, then cite that RO stability bounds the generalization gap. The short coming is that the theory only analyzes the discriminator, which do not seem much different compared to previous work analyzing classifiers. It would be much more interesting and novel to see an analysis of the joint learning process of generator and discriminator. Experiments: The experiments show a correlation between regularization, less information leakage and reduced generalization gap. In particular, to show information leakage, the authors propose a simple scheme of using the value of the discriminator output to decide if an image is from the training set. I am actually surprised that such a method could work. I think some of the claims regarding the experiments are a little strong, as their relationships are not necessarily causal. But of course, it is extremely hard, in almost every deep learning setup to establish strong causal relationship between change of modeling choices and change of performance. So I will not over criticize this point. The writing is generally easy to read. There might be a typo on line 249 where the \leq should be a \geq. --------------- After rebuttal Thank you for your response. I would like to maintain my score after the rebuttal, for the following reasons: I think the proposed theoretical improvement are hard to materialize. For example, it seems that composition of differential privacy will lead to a very loose bound, as information leakage will add up. I do not know if practical results can be obtained. I think overall the paper is of okay quality, the perspective is interesting, the writing is good. It's not the most novel or surprising paper, but I am happy to see this paper accepted at NeurIPS.

[Author Response · NeurIPS 2019]

We thank all reviewers for their valuable comments. In what follows, we will respond to their concerns one by one.

**To Reviewer #1**

**The author should evaluate different attack methods and show the experimental results.** Thanks for your advice.
Here we refer to the attack used in our paper as the threshold attack. In practice, we found that this simple threshold
attack suffices to detect some useful information of the training dataset and can validate the relationship between the
information leakage and the generalization ability of GANs. Thus, we mainly demonstrated the results of the threshold
attack. Actually, we have also shown the results of other attacks (i.e. black-box attack) in Section 4.3. We will highlight
these results in the revision.

**It is better to demonstrate some evaluations of the information leakage of Bayesian GAN.** Actually, we have
conducted attack experiments on the Bayesian GAN and found that Bayesian GAN is effective in reducing both the
information leakage and the generalization gap of GANs. We will add these results in the revision.

**To Reviewer #2**

**The pure theoretical advancement is good but not strong.** Thanks for your advice. Although the pure theoretical
part is somewhat straightforward, it demonstrates a novel and practical view of privacy protection to analyze the
generalization property. With the help of the properties of differential privacy, we have provided a more general and
concise proof of the uniform convergence in contrast to prior works. Moreover, we have mentioned that our theoretical
analysis can be extended to the case of joint learning of the generator and the discriminator (via the post-processing
property of the differential privacy [1]). This is useful to analyze the information leakage of both the generator and the
discriminator. We will enrich this part in the revision.

**Section 4 can be improved by providing guidelines of GAN design.** This is a good suggestion! The current results
have implicitly shown the connection between the information leakage and the generalization ability of GANs. This
empirical evidence indicates that we can improve the generalization ability of GANs by reducing the individual's
information leakage. This can be done by reducing each individual's influence on the final learned model. Thus we
can propose some design principles to reduce the information leakage via limiting each individual's influence, such as
clipping the per-sample gradient or adding Lipschitz conditions on both the generator and the discriminator. In the
revision, we will add a specific part to summarize these design principles.

**To Reviewer #3**

**Analysis of the joint learning process of generator and discriminator.** Thanks for your suggestion! We have
mentioned that the analysis of the joint learning process can be accomplished with the post-process property and the
composition theory of differential privacy (see line 158-161 in the paper). We will detail this part in the revision.

**The effectiveness of the attack method used in this paper.** Some recent works have shown that overfitting of a DNN
model is a sufficient condition for the success of the membership attack [2]. Thus, we infer that the success of the attack
in our paper comes from the overfitting of the discriminator, i.e. we can use the discriminator's output to distinguish the
training data from the test data.

**The typo.** Thanks for your careful review! We will correct it in the revision.

**More explanation of the practical impact, or more novel theoretical analysis.** This is good advice for improving
our paper! For the practical impact, our empirical evidence shows that we can easily reduce the information leakage
of the discriminator without the loss of the generated image's quality (via adding Lipschitz constraints). The results
of the black-box attack (in Table 2) also show that Lipschitz constraints can reduce the information leakage of the
generator (since the black-box attack depends on the information of the generator). This is meaningful in some
real-world applications, such as releasing the learned generator to the public for the research purpose. On the other
hand, with our theoretical insight, we can provide some design principles of GANs (see details in the second response to
the Reviewer # 2). We will add these potential impacts in the revision. For novel theoretical analysis, in the revision, we
can formulate the problem of the joint learning process and add the detailed analysis of this learning process (through
the composition theory of the generalization bound and differential privacy).

# References

[1] Cynthia Dwork and Aaron Roth. The algorithmic foundations of differential privacy. *FTTCS*, 2014.

[2] Long et al. Understanding membership inferences on well-generalized learning models. *CoRR*, 2018.


[Meta-Review · NeurIPS 2019]

All the reviewers liked the link between privacy and generative adversarial networks. The authors could also successfully answer the main concerns of the reviewers in their rebuttal.